# Health Behaviors and Health-Related Quality of Life in Female Medical Staff

**DOI:** 10.3390/ijerph19073896

**Published:** 2022-03-25

**Authors:** Maria Niestrój-Jaworska, Małgorzata Dębska-Janus, Jacek Polechoński, Rajmund Tomik

**Affiliations:** 1Department of Health-Related Physical Activity and Tourism, The Jerzy Kukuczka Academy of Physical Education in Katowice, 40-065 Katowice, Poland; m.niestroj-jaworska@awf.katowice.pl (M.N.-J.); r.tomik@awf.katowice.pl (R.T.); 2Institute of Sport Sciences, The Jerzy Kukuczka Academy of Physical Education in Katowice, 40-065 Katowice, Poland; j.polechonski@awf.katowice.pl

**Keywords:** quality of life, health behavior, medical staff

## Abstract

The aim of this study was to evaluate the relationship between the intensity of health behaviors and health-related quality of life in female mid-level medical staff. The study group consisted of 153 female mid-level medical staff members. The intensity of health behaviors was examined with the Polish version of Health Behavior Inventory. Health-related quality of life was verified with the Short Form Health Survey questionnaire (SF 36v2). Among the participants, 33% had low, 39% average, and 28% high intensity of health behaviors. The mental component of health-related quality of life was rated higher (83.3 ± 15.3 points) compared to the physical one. The lowest health-related quality of life was observed in the domain of “bodily pain”, while the highest was found for the domain of “social functioning”. Both the physical and mental components of health-related quality of life were significantly positively correlated with health behavior prevalence in all its categories. The post-hoc tests revealed the variation in physical and mental components of HRQoL according to the level of health behavior intensity.

## 1. Introduction

The dynamic increase in the incidence of chronic non-communicable diseases (NCDs) is a common occurrence in social welfare societies. Such conditions include metabolic diseases (obesity, type 2 diabetes) and diseases of the systems: cardiovascular, respiratory, skeletal, nervous, gastrointestinal, and some cancers [1]. It is accepted that chronic diseases and their consequences contribute not only to poor health but also to objective and subjective aspects of the quality of life of societies. 

Hence, research has identified the concept of health-related quality of life (HRQoL), defined by World Health Organization experts as an individual’s perception of his or her life position within the cultural context and value system in which he or she lives and in relation to his or her tasks, expectations, and standards set by environmental conditions [2]. Therefore, self-assessment of the quality of life is a comprehensive assessment of a person’s physical health, emotional state, independence in life, degree of independence from the environment, and personal beliefs and convictions. It complements the objective assessment of health with subjective feelings in accordance with the holistic understanding of health [3]. 

The quality of life as understood here is determined mainly by health status, which, according to the concept of health fields by Marc LaLonde, in over half of cases, depends on lifestyle [4], viewed as any forms of behavior (actions) manifested by people, which in the light of modern medical knowledge usually cause specific positive (healthy) or negative (unhealthy) effects [5,6]. 

Numerous publications have shown significant associations between the prevalence of health behavior and health status in various dimensions [6,7,8,9,10,11,12,13,14]. Researchers from the World Cancer Research Fund International [15] and the World Health Organization (WHO) [1] emphasize that physical activity of appropriate volume and intensity and a healthy diet are considered the most important positive health behaviors in preventive healthcare, as they are closely related to six of the seven factors of premature deaths [15]. At the same time, it has been scientifically confirmed that the health-promoting nature of these behaviors can determine the quality of life [16,17,18,19,20]. In studies conducted to date, the relationship between the prevalence of health behaviors and quality of life has been verified mainly in people of different age groups (adults and older adults), with different health status (healthy and chronically ill), while few studies have examined such relationships in specific professional groups [21,22]. 

Caring for the health of the public, and consequently improving to some extent the quality of life of society, is the responsibility of medical professionals. In addition to specialized medical knowledge acquired during professional training, they should be characterized by an above-average understanding of the importance of healthy lifestyles, and take positive actions in the area of primary and secondary prevention [5,23]. A high level of health awareness should foster a variety of health-promoting behaviors among health professionals. However, the results of the study indicate that healthy lifestyles are characteristic of a minority of medical professionals, which does not support the effectiveness of their promotion of health behavior [24,25,26].

The vast majority of mid-level medical staff (nurses, medical caregivers, and physiotherapists) comprise women [27]. The research conducted so far, aimed at assessing their health behavior and quality of life, has focused mainly on the professional group of nurses [28,29,30,31]. It turns out that the representatives of this group are generally characterized by a low intensity of health behaviors, while at the same time they are characterized by a high level of perceived job-stress. Consequently, the medical staff’s scores on quality of life in different countries are lower than those of the general population [32]. Research focuses mainly on verifying the relationship between the quality of life of the nursing group and their professional burnout, mental health, or life satisfaction [33,34,35]. There are no manuscripts aimed at assessing the dependence of the quality of life and multi-dimensional care for health expressed in different health behaviors in female medical staff. The research should also be extended to examine female workers from various professional groups of medical staff; not only nurses, who are investigated most often.

Therefore, the aim of this study was to evaluate the relationship between health behaviors of female mid-level medical staff and their health-related quality of life.

## 2. Materials and Methods

### 2.1. Characterization of the Study Participants

Female mid-level medical staff members (nurses, medical caregivers, physiotherapists) employed in 24-h health care facilities in southern Poland were invited to participate in the study. Inclusion criteria for the study were holding a position involving direct work with and care of patients and completing a full study program. Eventually, the study group consisted of 153 female mid-level medical staff members (age 41.27 ± 10.30 years, body weight 67.11 ± 10.86 kg, body height 164.33 ± 6.23 cm, BMI 24.88 ± 3.99 kg/m^2^, work experience 15.86 ± 11.78 years).

### 2.2. Research Methodology and Tools

The Polish version of the Short Form Health Survey questionnaire (SF 36v2) was used for the assessment of health-related quality of life (HRQoL). The questionnaire consists of 11 questions containing 36 statements that allow for defining two categories of quality of life: physical component summary (PCS) and mental component summary (MCS) and their eight domains [2,36]:−Physical functioning (PF);−Role-physical (RP);−Bodily pain (BP);−General health (GH); −Vitality (VT);−Social functioning (SF);−Role-emotional (RE);−Mental health (MH).

The subscales PF, RP, BP, and GH generate information about the physical category, while the other four subscales (VT, SF, RE, and MH) illustrate the mental component of health-related quality of life. Each response in each domain was assigned an appropriate score (ranging from 0 to 100 in each category), with the fewer points, the lower HRQoL The SF-36v2 questionnaire scores were calculated according to the algorithm of the Polish version of the questionnaire according to Żołnierczyk-Zreda et al. [2]. 

The prevalence of respondents’ health behaviors was verified using Juczynski’s Health Behavior Inventory (HBI), which consists of 24 questions on personal health-related behaviors [6,37]. The inventory evaluates the prevalence of health behaviors in four categories [37]:−Proper eating habits (PEH);−Pro-health activities (PhA);−Preventive actions (PA);−Positive mental attitude (PMA). 

Respondents rated how often per year they performed the health-related activities enumerated in the inventory on a five-point Likert scale, where: 1—almost never; 2—rarely; 3—from time to time; 4—often; 5—almost always. The health behavior index (HBI) was calculated using the questionnaire key. The raw scores are transformed into Standard Ten Scores scale (1–10 Sten), where 1–4 Sten means low scores, 5–6 Sten—average scores, and 7–10 Sten—high scores. It is assumed that the higher the health behavior intensity, the more potential health benefits in an individual [37]. HBI’s reliability is 0.85 for the entire inventory, whereas for individual subscales, it ranges from 0.60 to 0.65. For the current study, Cronbach’s alpha coefficients were: 0.80 for the entire scale, and from 0.65 to 0.80 for the subscales.

### 2.3. Research Procedure

Meetings with female mid-level medical staff members who were willing to participate were held before the main part of the study began. The purpose of the research, the procedure, and dates were discussed at the meeting. The survey was conducted in an auditory/individual format.

### 2.4. Ethics

The research procedure was approved by the Research Ethics Committee of the Medical University of Silesia in Katowice (PCN/0022/KB/277/19). All subjects were familiarized with the aim of the study prior to data collection. Participants took part in the research voluntarily and could discontinue their participation at any time. They provided written consent for the use of information collected during the examination.

### 2.5. Statistical Analysis

Descriptive statistics of variables (characteristics) such as numbers (*n*), fractions (%), arithmetic means ($\bar{x}$), standard deviations (SD), and minimum (Min) and maximum (Max) values were used in statistical analyses of the collected data. Spearman’s rank correlation coefficient (r_S_) was also used as a measure of the relationship between the variables. The Kruskal–Wallis test was used to compare the HRQoL in individuals with different levels of the health behavior index. The level of statistical significance was set at *p* < 0.05.

## 3. Results

The results of the HRQoL showed that the participants rated its mental component higher (83.3 ± 15.3 points) compared to the physical component (64.7 ± 10.1 points). For the physical component of quality of life, respondents rated the domain of “physical functioning” most favorably and “bodily pain” lowest. In the mental component, quality of life was found to be highest in the context of “social functioning” and lowest in the aspect of “role-emotional” (Table 1). 

Diagnosis of health behavior prevalence showed that the mean level of health behavior index in female medical staff members was 82.4 ± 13.2 points. Among the participants, 33% had low, 39% average, and 28% high rates of health behaviors. Female mid-level medical staff members were engaged most frequently in health behaviors from the category of “positive mental attitude” and least often in health behaviors from the category of “healthy practices” (Table 2).

Spearman’s rank correlation analysis showed significant positive relationships of the physical component summary and its two components: “bodily pain” and “general health” with the intensity of “proper eating habits”, “positive mental attitude”, and “pro-health activities”. Similar correlations were observed between the HRQoL’s domain of “physical functioning” and “proper eating habits” and pro-health activities (PhA) (Table 2).

The mental component summary increased with the increasing intensity of PEH, PMA, and PhA. The intensity of PA behaviors correlated positively with MCS and its component—MH (Table 2).

Based on the Kruskal–Wallis ANOVA test, there was significant variation in physical and mental components of health-related quality of life according to the level of health behavior index. Detailed post hoc analysis showed statistically significant differences (*p* < 0.05) between the low and average, as well as between the low and high index of health behaviors in five analyzed components of HRQoL (GH, SF, VT, MH, and MCS) Such relationships were also observed between the low and average HBI in three domains (PF, BP, and PCS), and between the low and high index in one domain (RE) (Figure 1).

## 4. Discussion

Assessing of the prevalence of health behaviors in female mid-level medical staff showed that it was high in only 27% of the respondents. Medical staff is generally characterized by a high level of health awareness, so one would expect a greater propensity for pro-healthy lifestyles, but it turns out that their level of health behavior does not differ significantly from that observed in the general population. This observation is also consistent with findings of studies conducted in the United States and Canada [38,39].

In our study, the least frequent area of health behavior was pro-health activities. This may be related to doing shift work, which, according to the findings of Peplonska et al. [40], by disorganizing the schedule of the standard day, contributes to reduced participation in physical activity, a tendency to snack, and frequent sleep deprivation. An increased tendency toward sedentary behavior and decreased participation in physical activity was observed in, e.g., a group of Italian nurses [41]. The results of our own study also showed an average prevalence of normal eating habits. It was also observed in Australian female mid-level medical staff [31]. This observation coincides with that made by Tsiga et al. who compared health behaviors of different professional groups in six European countries and showed that fast-food consumption among medical personnel was twice as high as that of other professionals [42].

Among the publications focused on assessing the quality of life of medical professionals, most studies examined the groups of nurses. The most frequently diagnosed area in this regard is the quality of work life, which, based on the results of the present study, is low [28,30]. Our study verified the health-related quality of life of mid-level medical staff. The results of the study showed that the HRQoL in women from this occupational group is above average. Similar results were obtained by Lewko et al. in the research of Polish nurses from the work experience study [34]. It turned out that the participants rated their quality of life significantly better for the category of mental health (MCS) compared to physical health (PCS). The opposite pattern was observed in female medical staff in Spain [35], Italy [33], and Greece [21]. The reason for such different findings may be the specificity of the study group. In the cited studies, a large part of the respondents were physicians, whose work is associated with very high stress [43], while in our study, we examined only female middle-level medical staff.

Our results showed significant positive correlations between HRQoL and the majority of HBI’s categories. Similar relationships have so far been demonstrated in both healthy [44,45,46,47,48,49] and sick [50,51,52,53] individuals of different ages and from diverse social backgrounds. While much of this research has examined the relationship of specific health behaviors with quality of life, our study diagnosed the overall prevalence of these behaviors. A study by Lee and Oh confirmed that physical activity and screening in a population of Korean older adults are significant predictors of quality of life [54]. Furthermore, the results published by Strine et al., who studied a systematic review and meta-analysis which the aim of the study was to quantitatively evaluate the relationship between health literacy (HL) and QoL, showed that unhealthy behaviors significantly reduce the quality of life [55]. 

In our study, we found that stronger correlations of HBI and HRQoL occurred for MCS and its domains. This observation is consistent with that made in a study of French adults by Tesier et al., who showed that increasing physical activity by 1 h per week results in a significant increase in HRQoL only in the MCS category [56]. A study by Wendel-Vos demonstrated similar relationships [57]. The reason for a stronger relationship of health behaviors with quality of life in the category of mental health observed in our study may be the specific nature of the work of mid-level medical staff, which is associated with increased physical exertion during the performance of professional duties. 

Both high HRQoL and care for health expressed in high HBI can contribute to better performance of professional duties and lower incidence of burnout in medical staff [58]. Employees who are more satisfied with life are characterized by higher commitment, optimism, and vigor, and perform their professional and social duties with greater accuracy and conscientiousness, often exceeding the level of expectations set by their superiors [59]. It is important to note that adequate involvement of nurses, medical caregivers, and physiotherapists in their work translates into the quality of care, satisfaction levels, and a sense of security of patients [60]. Therefore, it seems appropriate to conduct research aimed at developing programs to promote healthy attitudes in this professional group. 

Several limitations of this study should be mentioned. The study is preliminary in nature: it covered female middle-level medical staff from medical facilities in southern Poland. In further studies, it would be necessary to expand the research to include individuals from the medical facilities from other parts of the country. It is also planned to verify again the strength of the examined relationships considering potential confounders (e.g., age, income, number of working hours, social support at work, and work-related compensation strategies). It should also be emphasized that our study is based on self-reported data. It would be advisable to diagnose the health behaviors with the use of specific, objective methods of diagnosis and verify the relations examined in this study again.

## 5. Conclusions

Female mid-level medical staff rated their quality of life higher in the mental compared to the physical dimension. The lowest HRQoL was observed in the domain of “bodily pain”, while the highest was found for the domain of “social functioning.” The mean health behavior index of the female respondents was at the average level. Respondents were most likely to engage in health behaviors related to positive mental attitudes and least likely to engage in pro-health practices. Health-related quality of life in both the physical and mental components of quality of life was significantly positively correlated with health behavior prevalence in all its categories. Furthermore, stronger correlations of HBI and HRQoL for MCS and its domains were also found. Due to the fact that there is a significant positive correlation between health-related behaviors and the quality of life of medical personnel, it is necessary to implement information programs in this professional group, encouraging them to take appropriate behaviors and to expand health-related education in medical faculties.

## Figures and Tables

**Figure 1 ijerph-19-03896-f001:**
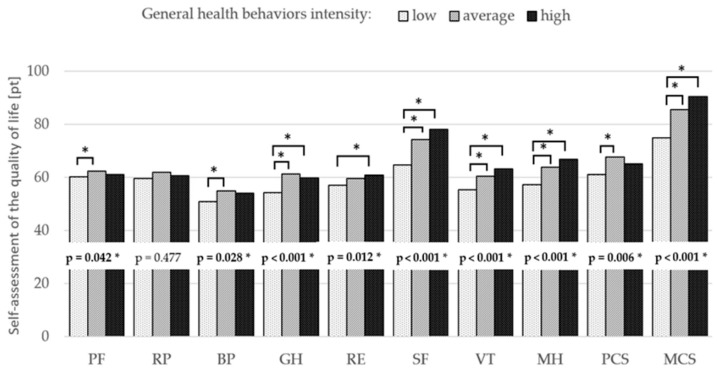
Health-related quality of life according to the level of health behavior inventory. Legend: *—significant difference (*p* < 0.05) within health behavior intensity, PF—physical functioning, RP—role-physical, BP—bodily pain, GH—general health, RE—role-emotional, SF—social functioning, VT—vitality, MH—mental health, PCS—physical component summary, MCS—mental component summary.

**Table 1 ijerph-19-03896-t001:** Health-related quality of life and the intensity of health behaviors in female medical staff.

**Heath Related Quality of Life**		
**Variable [pt]**	$\bar{x}$ **±****SD**	**Min-Max**
PF	61.2 ± 4.6	39.7–64.6
RP	60.8 ± 7.1	35.9–66.9
BP	53.3 ± 8.6	36.6–66.6
GH	58.5 ± 8.9	38.0–76.0
RE	59.1 ± 6.2	32.1–62.9
SF	72.2 ± 14.2	36.3–85.1
VT	59.4 ± 9.7	18.3–85.6
MH	62.4 ± 9.2	27.1–81.4
PCS	64.7 ± 10.1	34. 2–80.4
MCS	83.3 ± 15.3	21.4–97.1
**Heath Behavior**		
**Variable [pt]**	$\bar{x}$ **±****SD**	**Min-Max**
PEH	20.5 ± 4.9	6.0–30.0
PA	21.0 ± 4.2	9.0–30.0
PMA	21.6 ± 3.8	15.0–29.0
PhA	19.3 ± 3.6	11.0–28.0
HBI	82.4 ± 13.2	47.0–112.0

Legend: PF—physical functioning, RP—role-physical, BP—bodily pain, GH—general health, RE—role-emotional, SF—social functioning, VT—vitality, MH—mental health, PCS—physical component summary, MCS—mental component summary, PEH—proper eating habits, PA—preventive actions, PMA—positive mental attitude, PhA—pro-health activities, HBI—health behavior inventory.

**Table 2 ijerph-19-03896-t002:** Correlations of health behavior inventory with health-related quality of life in female respondents.

Variable	PEH	PA	PMA	PhA	HBI
	r_S_ *p*	r_S_ *p*	r_S_ *p*	r_S_ *p*	r_S_ *p*
PF	0.21 **0.009 ***	0.07 0.37	0.13 0.116	0.18 **0.026 ***	0.19 **0.016 ***
RP	0.05 0.516	−0.06 0.48	0.15 0.059	0.08 0.305	0.06 0.428
BP	0.2 **0.015 ***	0.15 0.067	0.26 **0.001 ***	0.25 **0.002 ***	0.25 **0.002 ***
GH	0.21 **0.009 ***	0.1 0.207	0.32 **<0.001 ***	0.3 **<0.001 ***	0.29 **<0.001 ***
RE	0.2 **0.015 ***	0.08 0.333	0.31 **<0.001 ***	0.25 **0.002 ***	0.026 **0.001 ***
SF	0.34 **<0.001 ***	0.16 0.054	0.4 **<0.001 ***	0.44 **<0.001 ***	0.24 **<0.001 ***
VT	0.22 **0.007 ***	0.16 0.052	0.43 **<0.001 ***	0.44 **<0.001 ***	0.37 **<0.001 ***
MH	0.28 **<0.001 ***	0.23 **0.004 ***	0.56 **<0.001 ***	0.39 **<0.001 ***	0.46 **<0.001 ***
PCS	0.21 **0.009 ***	0.08 0.303	0.27 **0.001 ***	0.27 **0.001 ***	0.25 **0.002 ***
MCS	0.32 **<0.001 ***	0.2 **0.011 ***	0.53 **<0.001 ***	0.46 **<0.001 ***	0.47 **<0.001 ***

Legend: PF—physical functioning, RP—role-physical, BP—bodily pain, GH—general health, RE—role-emotional, SF—social functioning, VT—vitality, MH—mental health, PCS—physical component summary, MCS—mental component summary, PEH—proper eating habits, PA—preventive actions, PMA—positive mental attitude, PhA—pro-health activities, HBI—health behavior inventory, r_S_—Spearman’s rank correlations, *—statistically significant correlation *p* < 0.05.

## Data Availability

The data presented in this study are available on request from the corresponding author.

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
