# Peer review of "Health Behaviors and Health-Related Quality of Life in Female Medical Staff"

_ijerph, 2022, doi:10.3390/ijerph19073896_

Round 1

Reviewer 1 Report

The article entitled Health Behaviors and Health-Related Quality of Life in Female Medical Staff is an interesting study of an important topic, which is the quality of life and health behavior of medical personnel, which turns out to be particularly important in the current conditions of overloading health care systems. The research used psychometric tools recognized in Polish cultural conditions to measure health behaviors and quality of life. The performed statistical analyzes are correctly selected. The results are interesting, presented clearly and aesthetically. A few minor notes to correct: - an editorial error crept in line 75: 41. {Citation} ± 10.30 years; - some of the cited literature is outdated (over 10 years), try to indicate newer references; - it would be worth referring to the obtained post-hoc results in the discussion. Thank you for the opportunity to review this interesting work, and I recommend that you accept it after minor corrections.

Author Response

We highly appreciate the detailed, valuable comments on our manuscript. The suggestions are very helpful for us, and we have incorporated them into the revised paper.

We have addressed all issues indicated in the review report, and we believe that the revised version meets the journal publication requirements.

In response to the Reviewer suggestions we responded to the below comments:

  1. An editorial error crept in line 75: 41. {Citation} ± 10.30 years.

Thank you for this comment. We have corrected this error.

  1. Some of the cited literature is outdated (over 10 years), try to indicate newer references;

According to this suggestion we have updated the literature.

  1. It would be worth referring to the obtained post-hoc results in the discussion.

The results of the analysis of variance are important in the research, while the values of the post-hoc tests are only informative, therefore they were not included in the discussion.

Reviewer 2 Report

The idea to evaluate the relationship between the intensity of health behaviors and health-related quality of life in female mid-level medical staff is a relevant issue. The manuscript is interesting and well written.  However, the present manuscript has some points worth to be mentioned in limitations, such as: small sample size and no male mid-level medical staff study participants.

In addition, I would like to invite the authors to correct following:

  1. line 74 -  “41.{Citation}±10.30” please correct
  2. line 90 - in my opinion references are in inappropriate place.
  3. line 94 – in the end of the sentence missing punctuation.
  4. line 104- reference is in inappropriate place.
  5. line 129- “collected material” please, replace with “collected data”
  6. line 143- “33% had low, 39% average, and 27% high rates” the sum is not equal to 100%, identical is in the abstract.
  7. pg. 4 – pay attention to abbreviations, there are several repetitions. For example line 150 -"pro-health activities " (PhA), on the same page line 155- "pro-health activities " (PhA).
  8. In the manuscript you report the results as “83.3±15.3”, but in the tables as “58,5±8,9”. Please, use for decimals “.” or “,” in the whole manuscript.
  9. line 174- “HRQOL” replace with “HRQoL”
  10. line 184 – “Diagnosis of the prevalence of health behaviors in female…” I think you are assessing and not diagnosing.
  11. line 250- “Health-related quality of life life” please, delete the second “life”

Author Response

We highly appreciate the detailed, valuable comments on our manuscript. The suggestions are very helpful for us, and we have incorporated them into the revised paper.

We have addressed all issues indicated in the review report, and we believe that the revised version meets the journal publication requirements.

In response to the Reviewer suggestions we responded to the below comments:

  1. line 74 -  “41.{Citation}±10.30” please correct

Thank you for this comment. We have corrected this error.

  1. line 90 - in my opinion references are in inappropriate place.

According to the suggestion, we have moved the citations elsewhere

  1. line 94 – in the end of the sentence missing punctuation.

We sincerely apologize for any omissions. We have supplemented it.

  1. line 104- reference is in inappropriate place.

Thank you for this comment. We have moved the citations elsewhere.

  1. line 129- “collected material” please, replace with “collected data”

Thank you for this comment. It was corrected as suggested.

  1. line 143- “33% had low, 39% average, and 27% high rates” the sum is not equal to 100%, identical is in the abstract.

Thank you for indicating this mistake. The fraction of students with high intensity of HB has been given incorrectly – instead of 27% it should be 28%. We corrected it  both in Abstract and in the first section of Results.

  1. pg. 4 – pay attention to abbreviations, there are several repetitions. For example line 150 -"pro-health activities " (PhA), on the same page line 155- "pro-health activities " (PhA).

We totally agree with your comment. Full names and abbreviations are used interchangeably in the revised version.

  1. In the manuscript you report the results as “83.3±15.3”, but in the tables as “58,5±8,9”. Please, use for decimals “.” or “,” in the whole manuscript.

Thank you for this suggestion. The record has been unified.

  1. line 174- “HRQOL” replace with “HRQoL”

We totally agree with your comment. It was replaced as suggested.

  1. line 184 – “Diagnosis of the prevalence of health behaviors in female…” I think you are assessing and not diagnosing.

We corrected the sentence according your suggestion.

  1. line 250- “Health-related quality of life life” please, delete the second “life”

We put the word “life” twice by mistake. It was corrected according to the suggestion.

Reviewer 3 Report

ijerph-1610443

This  manuscript entitled “Health Behaviors and Health-Related Quality of Life in Female 2 Medical Staff” aimed to evaluate the relationship between the intensity of health 10 behaviors and health-related quality of life in female mid-level medical staff.

The paper addresses an important topic regarding health in medical staff. Please find below my remarks that may be helpful in further improving the manuscript.Yet, the study remains a bit descriptive.

  1. What are the contributions beyond what is already known in health
    research on these and related issues?
    I assume that the references used on the paper should be more focus on the
  2. Female Medical Staff, instead of general population. If there is no studies in this particular population, it should be mentioned as a gap in the literature and also presented in the limitation sections. Studies considered in the discussion should approach similar populations.  
  3. Would you please clarify the gap in the literature and what this study adds?
  4. I may have missed but I did not see any information on the sample size. The sample needs to be described in more detail. Please provide calculation on the sample size.
  5. The paper is based on self-report data. Please discuss this important
    limitation in more detail.
  6. Confounders need to be considered (e.g., age, sex, income, number of
    working hours, social support at work, work-related compensation
    strategies, etc.).
  7. The conclusions need to be better elaborated. What are the conceptual
    implications on e.g. health behavior models? The authors may want to discuss in more detail the appliedimplications of the findings.
  8. What are the practical implications to support health in medical staff?

Author Response

We highly appreciate the detailed, valuable comments on our manuscript. The suggestions are very helpful for us, and we have incorporated them into the revised paper.

We have addressed all issues indicated in the review report, and we believe that the revised version meets the journal publication requirements.

In response to the Reviewer suggestions we responded to the below comments:

  1. What are the contributions beyond what is already known in health research on these and related issues? I assume that the references used on the paper should be more focus on the Female Medical Staff, instead of general population. If there is no studies in this particular population, it should be mentioned as a gap in the literature and also presented in the limitation sections. Studies considered in the discussion should approach similar populations.

We strongly agree with this suggestion. According to it we have added the paragraph describing the current state of art within  this topic (lines 66-78). We have also supplemented the Discussion section.

  1. Would you please clarify the gap in the literature and what this study adds?

According to the Reviewer’s suggestion we have clarified the gap in the current literature. Our study is  a preliminary one. We have examined the relations between health behaviors and quality of life in female mid-level medical staff. We have also extended the study group to different professional groups across medical staff, not only nurses which are the most frequently examined by researchers. (Lines 74-78).

  1. I may have missed but I did not see any information on the sample size. The sample needs to be described in more detail. Please provide calculation on the sample size.

The sample size is described in Materials and Methods section. “Eventually, the study group consisted of 153 female mid-level medical staff members (age 41±10,30 years, body weight 67,11±10,86 kg, body height 164,33±6,23 cm, BMI 24,88±3,99 kg/m2, work experience 15,86 ±11,78 years).”

  1. The paper is based on self-report data. Please discuss this important limitation in more detail.

We strongly agree with this comment. According to the suggestion, we have emphasized this limitation of our study at the end of Discussion section (Lines 265-267).

  1. Confounders need to be considered (e.g., age, sex, income, number of working hours, social support at work, work-related compensation strategies, etc.).

We are conscious that the relations between health-related quality of life and health behaviors should be verified in respect to the mentioned potential confouders. We have emphasized the preliminary character of our study in the manuscript. According to this we have not verified the above matters but we have described it as a limitation of our study. Additionaly our study group was rather homogenous – we have examined only women, employed in 24-hour health care centers, holding a position involving direct work with patients (Lines 85-87, 263-265).

6.The conclusions need to be better elaborated. What are the conceptual implications on e.g. health behavior models? The authors may want to discuss in more detail the appliedimplications of the findings.

Corrections were made and conclusions were supplemented.

7.What are the practical implications to support health in medical staff?

The conclusions were supplemented with practical implications.
